# Adaptive learning acceleration for nonlinear PDE solvers

**Vinicius Luiz Santos Silva**
Imperial College London
v.santos-silva19@imperial.ac.uk

**Pablo Salinas**
OpenGoSim
pablo.salinas@opengosim.com

**Claire E Heaney**
Imperial College London
c.heaney@imperial.ac.uk

**Matthew D Jackson**
Imperial College London
m.d.jackson@imperial.ac.uk

**Christopher C Pain**
Imperial College London
c.pain@imperial.ac.uk

## Abstract

We propose a novel type of nonlinear solver acceleration for systems of nonlinear partial differential equations (PDEs) that is based on online/adaptive learning. It is applied in the context of multiphase porous media flow. The presented method is built on four pillars: compaction of the training space using dimensionless numbers, offline training in a representative simplistic (two-dimensional) numerical model, control of the numerical relaxation (or other tuning parameter) of a classical nonlinear solver, and online learning to improve the machine learning model in run time (online training). The approach is capable of reducing the number of nonlinear iterations by dynamically adjusting one single global parameter (the relaxation factor) and by learning on-the-job the characteristics of each numerical model. Its implementation is simple and general. In this work, we have also identified the key dimensionless parameters required, compared the performance of different machine learning models, showed the reduction in the number of nonlinear iterations obtained by using the proposed approach in complex realistic (three-dimensional) models, and for the first time properly coupled a machine learning model into an open-source multiphase flow simulator achieving up to 85% reduction in computational time.

## 1   Introduction

The numerical solution of partial differential equations (PDEs) is an ubiquitous tool for modelling physical phenomena. Among PDE problems, multiphase flow in porous media is of paramount importance to understand, predict and manage subsurface reservoirs with applications to geothermal energy resources, $CO_2$ geological sequestration, hydrocarbon recovery, groundwater resources and magma reservoirs. However, the nonlinear nature of the problem and the strong coupling between the different equations make the numerical solution very challenging [Aziz, 1979, Jackson et al., 2015, Li and Tchelepi, 2015]. To solve the discretised nonlinear equations, Newton methods in fully implicit formulation have been used, although sequential methods have been gaining ground in recent years [Salinas et al., 2017a, Jiang and Tchelepi, 2019, Freitas et al., 2020]. Sequential methods are attractive because of their flexibility and extensibility, for example, each equation of the multiphysics problem can be solved separately through specialised solvers. The Picard iterative solver can work as a sequential approach, and can be seen as a sequential fixed-point iteration or a nonlinear block Gauss-Seidel

process [Silva et al., 2021]. This kind of method requires fewer conditions to achieve convergence than Newton methods [Elman et al., 2005, Lott et al., 2012], although Newton methods, in general, are faster than Picard iterative processes [Ortega and Rheinboldt, 1970, Jenny et al., 2006, Wong et al., 2019].

A great deal of effort has been devoted to accelerating sequential methods. To this end, the use of numerical relaxation (also known as backtracking) [Press et al., 2007] has shown promising results. Salinas et al. [2017a] proposed an acceleration technique based on numerical relaxation that is applied to the saturation field of multiphase porous media problems. Jiang and Tchelepi [2019] applied three nonlinear acceleration techniques to the sequential-implicit fixed point method. Numerical relaxation achieved the best overall performance compared to quasi-Newton and Anderson acceleration.

A promising approach in the field of machine learning comes in the form of learning from online/evolving data streams [Read et al., 2012, Kirkpatrick et al., 2017, Gomes et al., 2017a, Chen and Liu, 2018, Montiel et al., 2020, Hoi et al., 2021]. It can provide an attractive alternative to accelerate nonlinear PDE solvers, since in each numerical simulation several nonlinear iterations are performed, and they can be used to improve the convergence of the nonlinear solver. We can highlight two main branches of algorithms to train the machine learning model, instance-incremental methods [Cauwenberghs and Poggio, 2000, Oza and Russell, 2001, Losing et al., 2016, Hoi et al., 2021] and batch-incremental methods [Breiman, 1999, Polikar et al., 2001, Wang et al., 2003, Montiel et al., 2020]. Instance-incremental learning uses only one sample to update the machine learning model at each time, where batch-incremental learning uses a batch of multiple samples to update the model each time. Read et al. [2012] compared these two approaches for the task of classification of evolving data streams. They showed that both methods perform similarly given a limited resource, and that the optimal batch size depends on the problem in consideration.

In this work, we use an online learning approach to improve/update the machine learning model in run time, with the aim of continuously adapting to changes in the numerical PDE simulation. To the best of the authors' knowledge, this is the first work to use online learning to accelerate a numerical PDE solver. We list the contributions of our work as follows: (I) We propose an online/adaptive learning acceleration for nonlinear PDE solvers, and apply it to a broad class of multiphase porous media problems. It is able to control the convergence of the nonlinear solver by learning from previous nonlinear iterations of the running numerical simulation. (II) We perform a thorough investigation to select the best set of features (dimensionless numbers) used as input parameters for the proposed acceleration and compare different machine learning models for controlling the convergence of the solver. (III) For the first time, we fully integrate a machine learning model into the nonlinear solver of an open-source multiphase flow simulator, achieving up to 85% reduction in computational time. The source code, data and hardware configuration used in this work are available at `https://github.com/vlssanonymous/mlsolveracc`.

## 2   Related work

The success of machine learning in different fields has inspired recent applications with the aim of accelerating the convergence of the numerical PDE solutions. Greenfeld et al. [2019] proposed a new framework for multigrid PDE solvers, where a single mapping from discretization matrices to prolongation operators is learned using a neural network in an unsupervised learning procedure. Hsieh et al. [2019] proposed an approach to learn a fast iterative PDE solver tailored to a specific domain. It achieved significant speedups compared to standard formulations; however, it only works for linear solvers. Oladokun et al. [2020] used a random forest regression to determine the linear convergence tolerance of the nonlinear solver, and was able to reduce the number of linear iterations. Silva et al. [2021] proposed a machine learning approach to accelerate convergence of a nonlinear solver by dynamically controlling a relaxation parameter. The proposed approach was capable of reducing the number of nonlinear iterations, including models more complex than the training case. Kadeethum et al. [2022] used the prediction from a machine learning based reduced order model as a initial guess for the nonlinear solver iterations. Their approach maintains the full order model accuracy while accelerating the nonlinear solver convergence.

# 3 Governing equations and nonlinear solver

We report the formulation for incompressible flow with gravity and capillary pressure. Considering phase $\alpha$ of immiscible-fluid phases, the mass-balance equation is

$$\phi\frac{\partial S_\alpha}{\partial t}+\nabla\cdot\mathbf{q}_\alpha=Q_\alpha, \tag{1}$$

where $t$ is time. $\phi$ represents the rock porosity, $S_\alpha$ is the saturation of phase $\alpha$, $\mathbf{q}_\alpha$ is the Darcy velocity, and $Q_\alpha$ is a source term.

The multiphase Darcy's law for phase $\alpha$ is given by

$$\mathbf{q}_\alpha=\frac{k_{r\alpha}\mathbf{K}}{\mu_\alpha}(-\nabla p_\alpha+\rho_\alpha g\nabla z), \tag{2}$$

where $\mu_\alpha$, $k_{r\alpha}$, and $\mathbf{K}$ are the viscosity, relative permeability, and permeability tensor, respectively. $\rho$ is the density, $p$ is the pressure, $g$ is the gravitational acceleration, and $\nabla z$ is the gravity direction.

Considering a wetting $(w)$ and non-wetting $(nw)$ phase and including capillary pressure, $p_c$, the system of equations is closed by the constraints

$$p_c=p_{nw}-p_w, \tag{3}$$

$$S_w+S_{nw}=1. \tag{4}$$

The discretised form of the nonlinear system of equations formed by Eqs. (1), (2), (3) and (4) is solved by a Picard iterative method. It is implemented in the open-source code IC-FERST (Imperial College Finite Element Reservoir SimulaTor) [Gomes et al., 2017b, Salinas et al., 2017a,b, Obeysekara et al., 2021]. The solver comprises three main loops: the time loop, the coupling between saturation and pressure (nonlinear outer loop), and the coupling between saturation and velocity (nonlinear inner loop). Further details about the nonlinear solver can be found in Appendix A.

In order to accelerate the convergence of the nonlinear solver a relaxation parameter is used [Salinas et al., 2017a, Jiang and Tchelepi, 2019, Silva et al., 2021]. The method involves updating the saturation by weighting the new calculated field with the saturations obtained in previous inner loop iterations. However, an important question is how the relaxation parameter is chosen in each outer nonlinear iteration. A naive method could choose a static value for all outer iterations. Nevertheless, the relaxation parameters needs to be small enough to avoid divergence and as large as possible to accelerate convergence. The value of the relaxation factor plays an important role in the convergence of the nonlinear loops but it is not known *a-priori* and is problem specific. Moreover, even the optimal static value can result in more iterations than a suitable dynamic relaxation factor [Küttler and Wall, 2008, Jiang and Tchelepi, 2019, Silva et al., 2021].

# 4 Adaptive learning acceleration

The relaxation factor is of paramount importance to accelerate the convergence of nonlinear solvers. Therefore, we propose a novel approach to control the relaxation parameter in each outer nonlinear iteration, where we adapt the machine learning model to the changes in the numerical simulation during run time, as shown in Figure 1. We start with an offline part, where the machine model is trained using a dataset generated by a simple two-dimensional reservoir model. The machine learning model inputs are dimensionless numbers, simulation properties, and the number of inner nonlinear iterations at each outer nonlinear iteration. The output of the model is the value of the relaxation parameter. In this work, we test different sets of dimensionless numbers as input parameters, and we analyse several machine learning models in terms of accuracy, prediction time and simulation results.

After the offline stage, we use the machine learning model to determine the value of the numerical relaxation for a number of outer nonlinear iteration that we call $W$ (as in Figure 1). Following that, we can update the machine learning model using a batch-incremental method ($W > 1$), or an instance-incremental method ($W = 1$). After updating, we run more $W$ outer nonlinear iterations and update the model again. The process continues until the end of the numerical simulation. Given that batch-incremental and instance-incremental methods can perform similarly [Read et al., 2012], and to avoid the cost added to the numerical simulation each time we update the machine learning model,

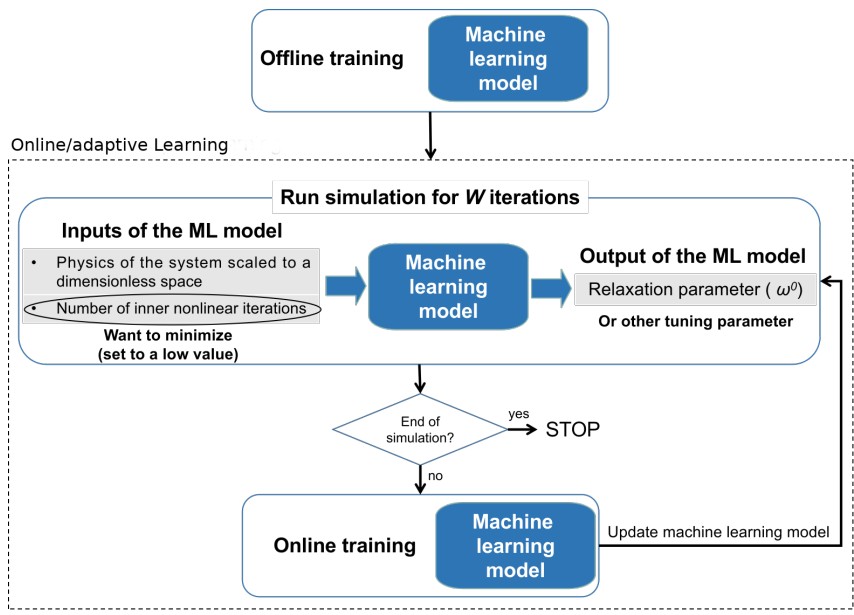

Figure 1: Adaptive learning acceleration for nonlinear PDE solvers.

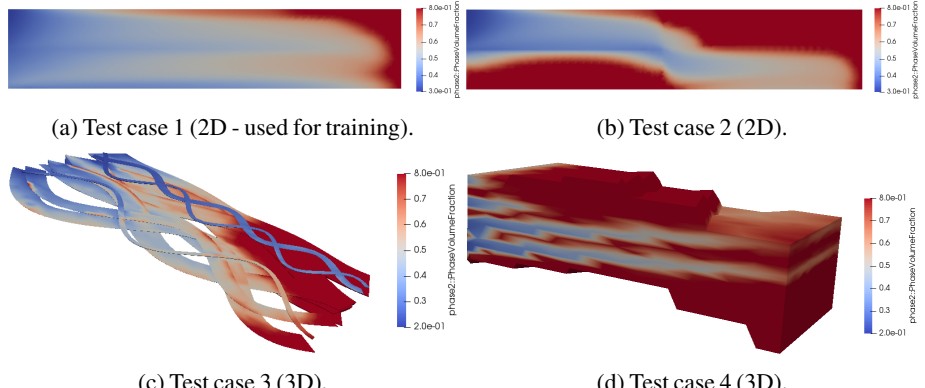

(a) Test case 1 (2D - used for training).

(b) Test case 2 (2D).

(c) Test case 3 (3D).

(d) Test case 4 (3D).

Figure 2: Saturation of the displaced phase in one point in time during the numerical simulation. Blue represents the injected phase and red the displaced phase. In all cases, we injected one phase on the left and produce both phases on the right.

we choose to use batch-incremental methods here (or $W > 1$). We also test different batch sizes in order to determine the best one for the multiphase porous media problem. It is worth mentioning that since we have a fixed simulation time the number of updates of the machine learning model is finite, different from classical online learning problems.

The main goal of the adaptive learning acceleration is to train the machine learning model offline using a simple reservoir model (that is fast to run). Then apply the proposed acceleration to more complex and challenge reservoirs, while still learning from them in run time. During the numerical simulation, we learn from previous nonlinear iterations of the running reservoir model in order to better calculate the relaxation factor for the coming iterations. Further details about the input parameters and the offline training can be found in Appendixes B and C, respectively.

## 5 Numerical results

The adaptive learning acceleration is tested in four reservoir cases. All test cases represent two phase immiscible flow in a porous medium. The first is the two-dimensional layered reservoir model used for training. The second is a two-dimensional heterogeneous reservoir model with permeability varying

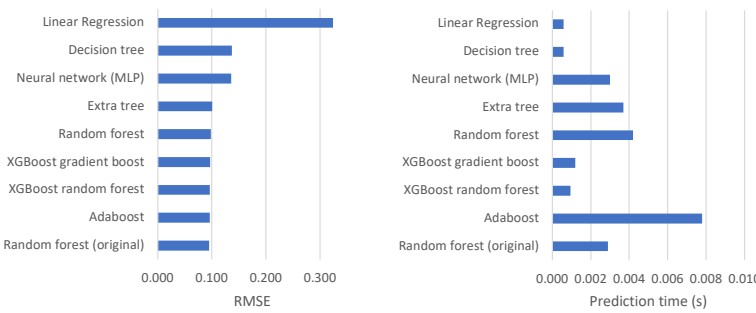

Figure 3: Comparison of different machine learning models. (left) The test root mean square error. (right) The one-instance prediction time

in four quadrants. The third is a realistic three-dimensional channelized reservoir model, and the fourth is a realistic three-dimensional faulted reservoir model. All reservoir models were tested with gravity and capillary pressure. Figure 2 shows the saturation map for all cases. We inject one fluid on the left and produce both fluids on the right. More details about the four numerical model can be found in the Appendix D.

In the following sections, we analyse and select the best machine learning models in terms of accuracy, prediction time and simulation results (we also select the best set of dimensionless numbers as input parameters, for more details see Appendix E). Following that, we apply and analyse the proposed adaptive learning acceleration. Finally, we integrate the best machine learning model into IC-FERST in order to actually generate a walltime reduction. As a baseline for comparison, we use here the nonlinear solver acceleration presented by Silva et al. [2021]. They propose to use a random forest to calculate the relaxation parameter, but using only an offline training. For the remainder of this paper, the random forest model used in Silva et al. [2021] will be also referred as original random forest.

## 5.1 Machine Learning model selection

We compare several machine learning models for controlling the numerical relaxation. All of them were implemented using Pedregosa et al. [2011], except for the xgboost models where we used Chen and Guestrin [2016] and the neural network (multilayer perceptron - MLP) where we use Abadi et al. [2015]. The input in all the models are the 17 features (Set 1) selected in Appendix E, except for the original random forest that uses the 38 features in Table 4. For each model we run a hyperparameter optimization using a grid search with a 3-fold cross-validation. Again, for the sake of comparison, the results reported in this section only consider the offline training. Figure 3 shows a comparison of the different models. In Figure 3 (left) we can see the test RMSE of the machine learning models. Apart from the linear regression all the models present similar RMSE. Figure 3 (right) shows the prediction time for one instance. For each model, we run the prediction 1000 times and average the resulted prediction time. As we use the machine learning model to generate the relaxation parameter in each outer nonlinear iteration, we want the prediction time to be as low as possible. We can notice that the xgboost models, linear regression and decision tree exhibit the lowest prediction times.

Figure 4 shows the reduction in the number of nonlinear iterations (improvement) for all the machine learning models, except for the adaboost and decision tree. This is because test case 4 failed to converge when using them. The models that generate better results than the original random forest are the xgboost random forest and the random forest (both using the Set 1). The cumulative number of nonlinear iterations during the numerical simulation for all the machine learning models and the case with no relaxation can be seen at Appendix F.

## 5.2 Online learning

In the previous section, we focused on the offline training. The two machine learning models that generated better results than the original random forest are the xgboost random forest and the random forest. Adding that ensemble learners have the advantage of being flexible, as new learners can be selectively added or removed, that they are usually easier to optimize, and that they are often used when learning

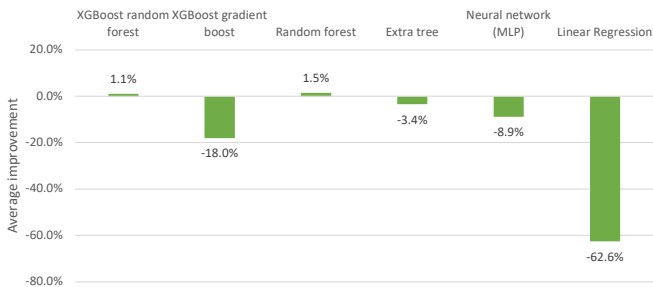

Figure 4: Comparison of different machine learning models. Average improvement in the number of nonlinear iterations compared with the original random forest.

Table 1: Adaptive learning acceleration using the random forest, and a bagging strategy for the online learning. Improvement (reduction) in the number on nonlinear iterations compared with the original random forest.

| TEST CASE | RF | $W$=25 | $W$=50 | $W$=200 |
|---|---|---|---|---|
| CASE 1 | 2.1% | 2.8% | 3.7% | 1.2% |
| CASE 2 | 0.0% | -1.8% | 0.6% | 0.5% |
| CASE 3 | -0.4% | 2.7% | 0.9% | 2.7% |
| CASE 4 | 4.3% | 5.0% | 4.5% | 4.2% |
| AVG. REDUCTION | 1.5% | 2.2% | 2.4% | 2.1% |

Table 2: Adaptive learning acceleration using the xgboost random forest, and a boosting strategy for the online learning. Improvement (reduction) in the number on nonlinear iterations compared with the original random forest.

| TEST CASE | XGBRF | $W$=25 | $W$=50 | $W$=200 |
|---|---|---|---|---|
| CASE 1 | 0.5% | -0.4% | 2.4% | 0.8% |
| CASE 2 | -0.6% | -1.3% | 0.0% | 0.0% |
| CASE 3 | 1.8% | -1.3% | 0.4% | 0.4% |
| CASE 4 | 2.8% | 3.4% | 3.3% | 2.9% |
| AVG. REDUCTION | 1.1% | 0.1% | 1.5% | 1.0% |

from evolving data streams [Oza and Russell, 2001, Read et al., 2012, Gomes et al., 2017a, Montiel et al., 2020], we have selected these two models to test the online/adaptive learning shown in Figure 1.

Many state-of-the-art ensemble methods for data stream learning are adapted versions of bagging and boosting strategies [Gomes et al., 2017a, Montiel et al., 2020]. In this work, given that the XGBoost library [Chen and Guestrin, 2016] already supports a boosting strategy for training continuation, we apply the online learning method Adaptive XGBoost, proposed by Montiel et al. [2020], to the xgboost model. Nonetheless, we consider here that we already have the first learner trained offline. In the online stage, as new outer nonlinear iterations arrive, they are stored in a buffer of size $W$. Once the buffer is full, we train a new member of the ensemble with the residuals from the previous members. It is worth noting that we do not reach the maximum ensemble size, unlike Montiel et al. [2020], because the time of the numerical simulation is finite and consequently the number of updates (usually less than 10 updates by numerical simulation). For the random forest, we modify the method proposed by Montiel et al. [2020] to work as a bagging strategy, since this is the strategy supported by the random forest in the Scikit-learn library [Pedregosa et al., 2011]. As in the Adaptive XGBoost method, as new outer nonlinear iterations arrives, they are stored in a buffer of size $W$. The difference in the random forest is that once the buffer is full, we train a new set of ensemble members with the samples in the buffer, and add it to the previous ensemble. The hyperparameters used to configure this methods are described in Appendix G.

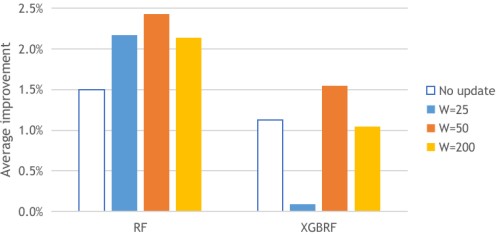

Figure 5: Average improvement (reduction) in the number of nonlinear iterations compared with the original random forest. The Base case is considering no online updating.

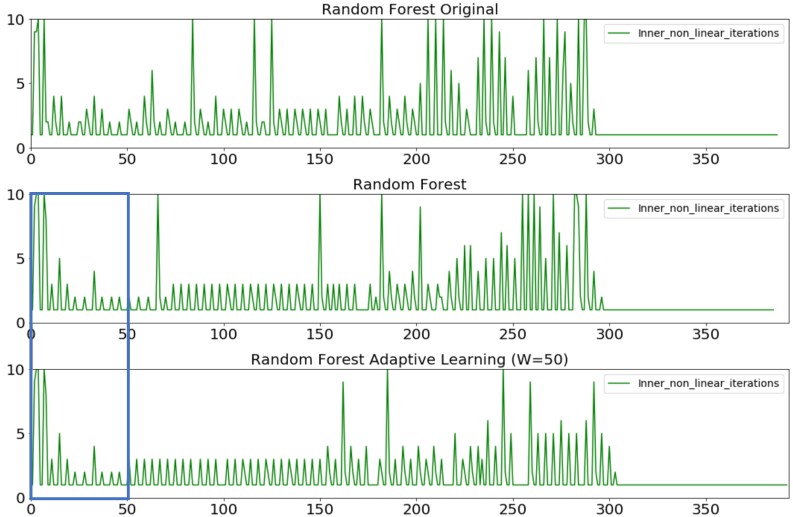

Figure 6: Comparison of the number of inner nonlinear iteration over the simulation period, for test case 1. The horizontal axes are the outer nonlinear iterations and the vertical axes the number of inner nonlinear iterations.

We applied these two strategies to the four test cases shown in Figure 2. Tables 1 and 2 show the percentage reduction (improvement) in the number of nonlinear iterations compared with the original random forest (state-of-the-art results from Silva et al. [2021]). The columns RF and XGBRF represent the two machine learning models selected from the previous section, but without the online learning stage. In the batch-incremental learning, the size of the batch must be chosen to provide a balance between accuracy (large batches) and response to new instances (smaller batches) [Read et al., 2012]. In this context, we test different buffer sizes ($W$) represented in the remaining columns of Tables 1 and 2. Figure 5 also shows the average reduction of each one of these strategies. We also notice that the strategy with buffer size equals 50 ($W = 50$) produced the best average performance for both machine learning models. Also for the strategy with buffer size equals 25 ($W = 25$), the number of nonlinear iterations became worse than the original random forest for some test cases. This can indicate that updating the machine learning model with short buffer sizes does not necessarily improve the final results.

Figure 6 shows the comparison of the number of inner nonlinear iterations over the simulation period, for the test case 1. The horizontal axes represent the outer nonlinear iterations. We can notice from the blue box on the left, that for the first 50 outer nonlinear iterations the results from the random forest with the online/adaptive learning and the one without it are the same. This is because the buffer has size 50 ($W = 50$), then until it is full no update is performed in the machine learning model. We can also see a clear reduction in the inner nonlinear iterations from the original random forest to the online/adaptive learning. It is worth noting that approximately after 300 outer nonlinear iterations the shock front (the interface between the two fluids in the porous media) has passed the simulation domain, making the nonlinear solver convergence much easier for the three strategies, only one inner nonlinear iteration is required to reach the convergence criteria.

Table 3: Reduction in run time and number of nonlinear iterations. Comparison between the proposed method (with no online update) and the default dynamic relaxation presented in IC-FERST [Salinas et al., 2017a].

| CASE | RUN TIME | NONLINEAR ITERATIONS |
|---|---|---|
| TEST CASE 1 | 19% | 13% |
| TEST CASE 2 | 34% | 37% |
| TEST CASE 3 | 9% | 10% |
| TEST CASE 4 | 85% | 81% |

## 5.3 Direct coupling

All of the previous models, including the original random forest [Silva et al., 2021], were not properly coupled with the numerical simulator in Fortran. An external Python script was called in each outer nonlinear iteration to generate the relaxation factor and it was passed "on the fly" to the numerical simulator. For that reason, no actual reduction in walltime was observed. Analysing the prediction time (Figure 3), the reduction in the number of nonlinear iterations (Figure 4), and considering the viability to integrate the machine learning code with the numerical simulator (IC-FERST) in Fortran, we chose the xgboost random forest to perform the coupling. A Fortran API for the xgboost was developed (available at `https://github.com/vlssanonymous/mlsolveracc/tree/master/xgboost_coupling`), and we were able to load the machine learning model in memory and call it in each outer nonlinear iteration using a Fortran code. Although, the online training of the machine learning model was not yet supported in the API, thus the results in this section consider $W = \infty$, which means no online update (the online update would further improve the results). Table 3 shows the reduction in the number of nonlinear iterations and walltime, when we compare the proposed method with the default dynamic relaxation presented in IC-FERST [Salinas et al., 2017a]. It is a dynamic relaxation that adjusts the relaxation parameter mainly based on the CFL number. It has demonstrated great improvements in the convergence rate of the solver for a wide range of cases [Salinas et al., 2017a], and up to this point was the best available approach to calculate the relaxation in IC-FERST. The method proposed here is able to outperform this approach, showing actual run time improvements. The results in table 3 show that the reduction in walltime is, in some cases, even greater than the reduction in the number of nonlinear iterations. It means that for some cases, the proposed method not only reduces the number of nonlinear iterations, but also makes faster the solution of each nonlinear iteration.

## 6 Conclusion

We propose a robust and efficient online/adaptive learning acceleration for nonlinear PDE solvers. The proposed approach was applied to complex/realistic two and three-dimentional subsurface reservoirs and was capable of reducing the number of nonlinear iterations without compromising on the accuracy of the results. We also present a thorough analysis of the performance of the presented acceleration technique. This analysis includes the use of different machine learning models, different dimensionless parameters, and different online learning strategies. In this study, we were able to select the most appropriate number of features (dimensionless numbers) used as inputs, and the most suitable machine learning models in terms of accuracy, prediction time and simulation results. Furthermore, for the first time we integrated a machine learning model into the nonlinear solver of an open-source, multiphase flow simulator (IC-FERST) and applied it to a set of challenging multiphase porous media flow problems. We were able to reduce simulation time by 37% on average, and up to 85% in the best case. Compared to other approaches our method is simple to implement and does not require retraining the machine learning model (offline training) when applied to other simulation domains or sets of parameters. Furthermore, not only does it learn offline, but also learns during the simulation to which it is applied. We believe that the online/adaptive learning acceleration is not limited to the multiphase porous media flow. It is applicable to any other system for which a relaxation technique (or other tuning parameter) can be used to stabilise the nonlinear solver.

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

## A    Nonlinear solver

We use a Picard iterative method to solve the discretised form of the nonlinear system of equations formed by Eqs. (1), (2), (3) and (4). Figure 7 describes the overall method. The solver comprises three main loops. The solid line represents the time loop, the dotted line denotes the coupling between saturation and pressure (nonlinear outer loop), and the dashed line iterates over saturation and velocity (nonlinear inner loop). In the inner loop, after calculating the saturation the velocity is updated. A new saturation is estimated using the new velocity. The process continues until convergence or the maximum number of iterations are reached. Following that, a new pressure is estimated in the outer loop based on the new saturation, and a new velocity is calculated from the pressure estimation. Then, a new inner loop iteration starts. We repeat the process until convergence or the maximum number of iterations is reached.

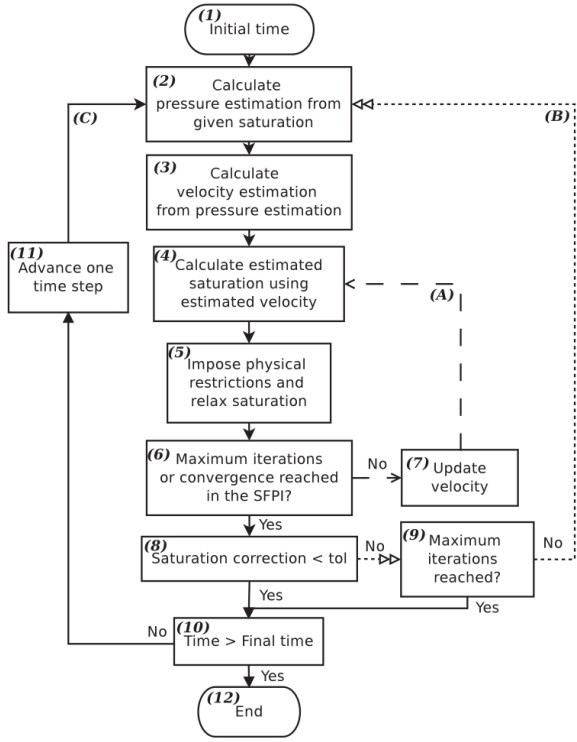

Figure 7: Flow chart of the nonlinear solver iterations. Reprinted from [Salinas et al., 2017a].

In order to accelerate the convergence of the nonlinear solver a relaxation parameter is used [Salinas et al., 2017a, Jiang and Tchelepi, 2019, Silva et al., 2021]. The method involves updating the saturation by weighting the new calculated field with the saturations obtained in previous inner loop iterations. The new saturation is calculated as

$$S^k = \omega^k \tilde{S}^k + (1-\omega^k)S^{k-1} + (1-\omega^k)^{\beta+1}\omega^k(S^{k-2}-S^{k-1}), \tag{5}$$

where $S^k$ is the new saturation after relaxation, $\tilde{S}^k$ is the saturation obtained after $(4)$ in Figure 7, and $k$ is the index representing the inner-loop current iteration. $\omega$ is the relaxation parameter, and $\beta$ is the exponent that controls the relative importance of $S^{k-1}$ and $S^{k-2}$. In this work, $\beta$ is considered constant with the value of $0.4$ as in Salinas et al. [2017a].

For each outer nonlinear iteration, having the initial value of $w^0$ the subsequent relaxation parameters ($w^k$) are calculated to yield the best convergence ratio reducing the residual of the saturation [Salinas et al., 2017a]. An important question is how $w^0$ is chosen in each outer nonlinear iteration. A naive method could choose a static value for all outer iterations. Nevertheless, the relaxation parameters needs to be small enough to avoid divergence and as large as possible to accelerate convergence. The value of $w^0$ plays an important role in the convergence of the nonlinear loops but it is not known *a-priori*

and is problem specific. Moreover, even the optimal static value can result in more iterations than a suitable dynamic relaxation factor [Küttler and Wall, 2008, Jiang and Tchelepi, 2019, Silva et al., 2021].

## B    Input parameters and dimensionless numbers

The combination of the viscous, capillary and gravitational forces drives the spatial distribution of fluids during multiphase flow in porous media [Hoteit and Firoozabadi, 2008, Li and Tchelepi, 2015, Debbabi et al., 2017a,b, 2018b]. The relative importance of these mechanisms depends on the combination of the fluid rock properties, the system length-scales and flow rates [Debbabi et al., 2018b]. For that reason, we choose the inputs of the machine learning model to be the dimensionless numbers presented on Debbabi et al. [2017a,b, 2018a]. Using dimensionless parameters to train and control the machine learning acceleration allow us to use a simple two-dimensional layered reservoir for training, while also exploring a large portion of the physical space. Table 4 summarizes all the input parameters.

Table 4: Input parameters for the machine learning model. CFL stands for Courant–Friedrichs–Lewy (CFL). The features where MIN, MAX and AVERAGE applies are defined control-volume wise.

| FEATURES | SILVA ET AL. [2021] | SET 1 | SET 2 |
|---|---|---|---|
| EFFECTIVE ASPECT RATIO | ONE VALUE | ONE VALUE | ONE VALUE |
| DARCY VELOCITY | AVERAGE, MAX AND MIN | AVERAGE | AVERAGE |
| TOTAL MOBILITY | AVERAGE, MAX AND MIN | AVERAGE | AVERAGE |
| CFL NUMBER | MAX VALUE | MAX VALUE | MAX VALUE |
| SHOCK-FRONT CFL NUMBER | MAX VALUE | MAX VALUE | MAX VALUE |
| SHOCK-FRONT NUMBER RATIO | ONE VALUE | ONE VALUE | ONE VALUE |
| SHOCK-FRONT MOBILITY RATIO | AVERAGE, MAX AND MIN | AVERAGE | AVERAGE |
| LONGITUDINAL CAPILLARY | AVERAGE, MAX AND MIN | AVERAGE | AVERAGE |
| TRANSVERSE CAPILLARY | AVERAGE, MAX AND MIN | AVERAGE | × |
| BUOYANCY NUMBER | AVERAGE, MAX AND MIN | AVERAGE | × |
| LONGITUDINAL BUOYANCY | AVERAGE, MAX AND MIN | AVERAGE | AVERAGE |
| TRANSVERSE BUOYANCY | AVERAGE, MAX AND MIN | AVERAGE | × |
| VANISHING ARTIFICIAL DIFFUSION | AVERAGE, MAX AND MIN | AVERAGE | AVERAGE |
| TRANSPORT EQUATION RESIDUAL | ONE VALUE | ONE VALUE | ONE VALUE |
| NUMBER OF FEATURES: | 38 | 17 | 14 |

## C    Training dataset

We run the offline training using results from a multiphase flow simulation in a simple two-dimensional layered model, as shown in Figure 8. We inject one phase on the left side and produce both phases on the right side. The aim is for the injected phase to push the displaced phase through the porous media. Because the reservoir model must be run numerous times to generate the dataset for training, employing a simple reservoir model considerably simplifies the training process.

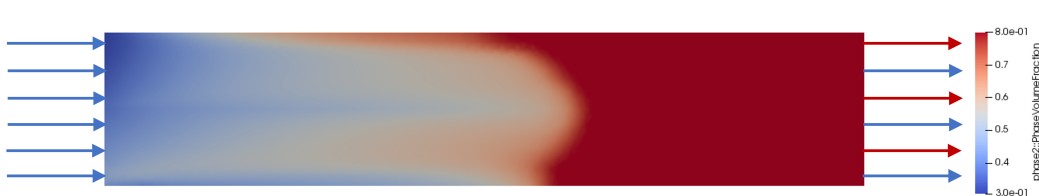

Figure 8: Two-dimensional homogeneous layered reservoir used for training. It shows the saturation of the displaced phase in one point in time during the numerical simulation. Blue represents the injected phase and red the displaced phase.

In order to perturb the physical parameters presented in Table 4 for generating the offline training set, we generate 6500 simulations changing the following simulations inputs: porosities, horizontal and

vertical permeabilities, time step, relative permeability, capillary pressure, viscosity, gravity magnitude and direction, and the relaxation factor. Figure 9 shows the learning curve for one of the machine learning models tested (a random forest). The dataset comprises 6500 simulations, that represents nearly $1.8 \times 10^6$ instances (outer nonlinear iterations), and were used for the training (80%) and test set (20%). In Figure 9, it can be observed that the rate at which errors decrease becomes marginal once we surpass this number of instances. Therefore, the inclusion of additional training simulations would not lead to a substantial reduction in errors.

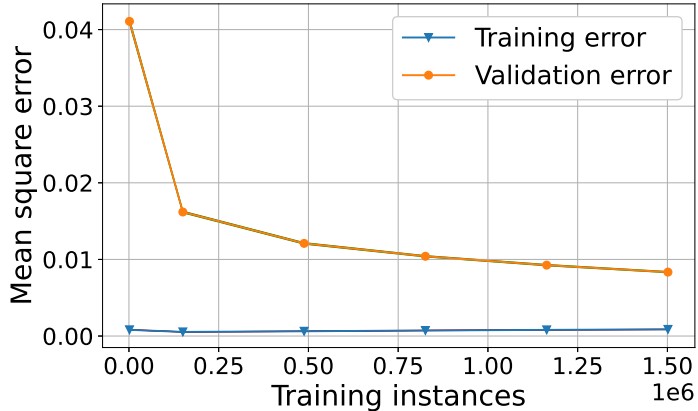

Figure 9: Learning curve. The vertical axis is the error of the machine learning model, and the horizontal axis represents the size of the dataset used for training.

In the online stage (online/adaptive learning), for each outer nonlinear iteration we store: the dimensionless number in Table 4, the relaxation factor used, and the number of inner nonlinear iterations performed. After running $W$ outer nonlinear iterations, we then run the online training. We repeat this process every $W$ nonlinear iterations until the end of the numerical simulation.

## D   Test case description

We evaluated the adaptive learning acceleration across four distinct reservoir models. Each of them represents a two-phase immiscible flow within a porous medium:

1. Test case 1: two-dimensional layered reservoir model used for training.

2. Test case 2: heterogeneous two-dimensional model with distinct permeabilities across its four quadrants.

3. Test case 3: more intricate three-dimensional model simulating channelized formations.

4. Test case 4: complex three-dimensional model with faulted structures.

For all models, both gravity and capillary pressure were considered. Figure 2 provides a saturation map for each case. The capillary pressure and relative permeability were modeled using the Brooks–Corey framework Brooks and Corey [1964]. In our tests, we injected one phase from the left and extracted both phases from the right.

We use as convergence criteria for the nonlinear solver that the relative mass conservation of the system has to be below $10^{-3}$ and the infinite norm of the saturation difference between two consecutive nonlinear iterations has to be below $10^{-2}$, within a time step. Furthermore, the maximum thresholds for the outer and inner nonlinear iterations are 30 and 10, respectively. When contrasting different scenarios, we calculate the total number of nonlinear iterations as the sum of the outer iterations and a third of the inner iterations. This method is consistent with the approaches described by Salinas et al. [2017a] and Silva et al. [2021].

### D.1 Test case 1

Test case 1 is the same reservoir model used to generate the dataset for the offline training. It is a two-dimensional model with two layers (Figure 10). The porosity in the top layer is 10% and the horizontal permeability is 200 mD. The bottom layer has 20% of porosity and 100 mD of horizontal permeability. The vertical permeability is equal to $10\%$ of the horizontal. The viscosity ratio between the injected and the displaced fluid is 5, the density contrast between phases is 300 $kg/m^3$, and the entry capillary pressure is 1000 Pa.

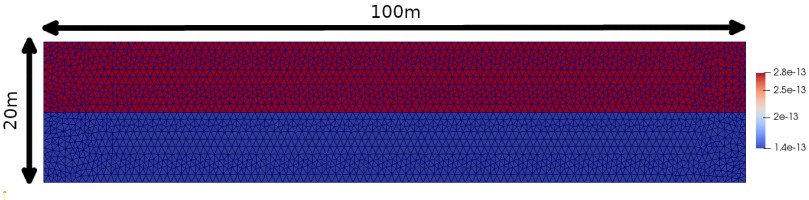

Figure 10: Test case 1 permeability field in m$^2$.

### D.2 Test case 2

Test case 2 is a two-dimensional reservoir with four domains arranged as quadrants (Figure 11). The permeabilities in the top layer are 200 mD and 20 mD, whereas the permeabilities in the bottom layer are 10 mD and 100 mD. The vertical permeability is equal to the horizontal and the porosity is 20% in all layers. The viscosity ratio between the displaced and the injected fluid is 5, the density contrast between phases is 300 $kg/m^3$, and the entry capillary pressure is 1000 Pa.

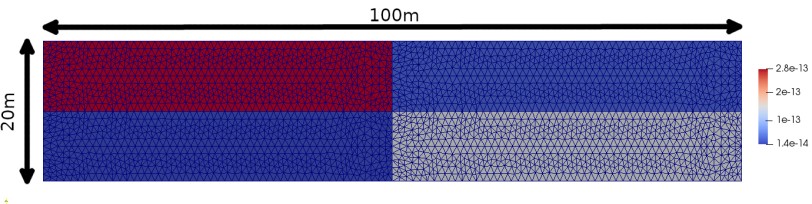

Figure 11: Test case 2 permeability field in m$^2$.

### D.3 Test case 3

A more realistic three-dimensional model of fluvial sandstone channels embedded in a low permeability mudstone background is also tested (Figure 12). The channels are divided in three sets, thin channels, medium size channels, and wide channels. The permeabilities in each set of channels are 1000 mD, 200 mD, and 100 mD, respectively. The vertical permeability is equal to the horizontal and the porosity is 20% in all channels. The viscosity ratio between the displaced and the injected fluid is 5, the density contrast between phases is 289 $kg/m^3$, and the entry capillary pressure is 100 Pa.

### D.4 Test case 4

Test case 4 is a second more realistic three-dimensional model of a faulted reservoir (Figure 13). The reservoir comprises a sequence of alternated sandstone and mudstone layers. The porosity in the sandstone layers is 10% and the horizontal permeability is 1000 mD. The mudstone layers have 20% porosity and 1 mD of horizontal permeability. The vertical permeability is equal to the horizontal. The viscosity ratio between the displaced and the injected fluid is 5, the density contrast between phases is 300 $kg/m^3$, and the entry capillary pressure is 10000 Pa.

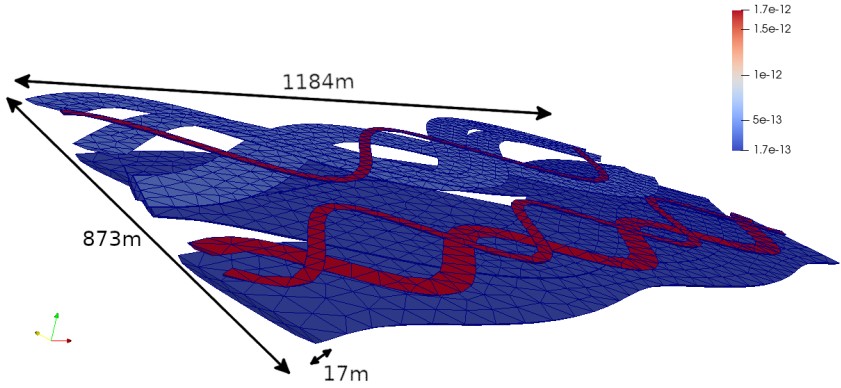

Figure 12: Test case 3 permeability field in m$^2$.

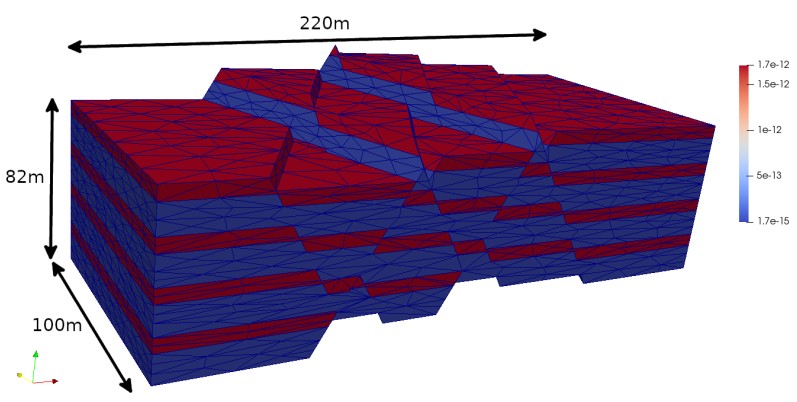

Figure 13: Test case 4 permeability field in m$^2$.

## E    Machine learning feature selection

We investigate different sets of features, in order to evaluate their effect on the proposed method. In Silva et al. [2021] a set of 38 features were chosen, and among them the average, minimum and maximum were calculated for the parameters defined control-volume wise, excepted for the Courant–Friedrichs–Lewy (CFL) number where it is already known that the maximum value plays an important role in the nonlinear solver convergence [Salinas et al., 2017a, Obeysekara et al., 2021]. In this work, we test two new sets of features as shown in Table 4. We notice that the average, minimum and maximum values are highly correlated when considering the same parameter. Therefore, in the first set (Set 1) we consider only the average. For the second set (Set 2), we have looked the feature importance of the original random forest, and for the capillary and buoyancy numbers we included only the most important features. As for the original random forest, we run a hyperparameter optimization using a grid search with a 3-fold cross-validation for the two new sets of features. The implementation of the random forests used here is the one in Pedregosa et al. [2011]. For the sake of comparison, the results reported in this section only consider the offline training.

Figure 14 shows a comparison of the different number of features. In Figure 14a we can see the test root mean square error (RMSE) of the random forest models for the three sets of features. No significant difference in the RMSE can be spotted when comparing the models. However, in Figures 14b and 14c we can notice that the reduction in the number of nonlinear iterations (improvement) is greater for Set 1 (fewer features). Set 2 (many fewer features) increased the average number of nonlinear

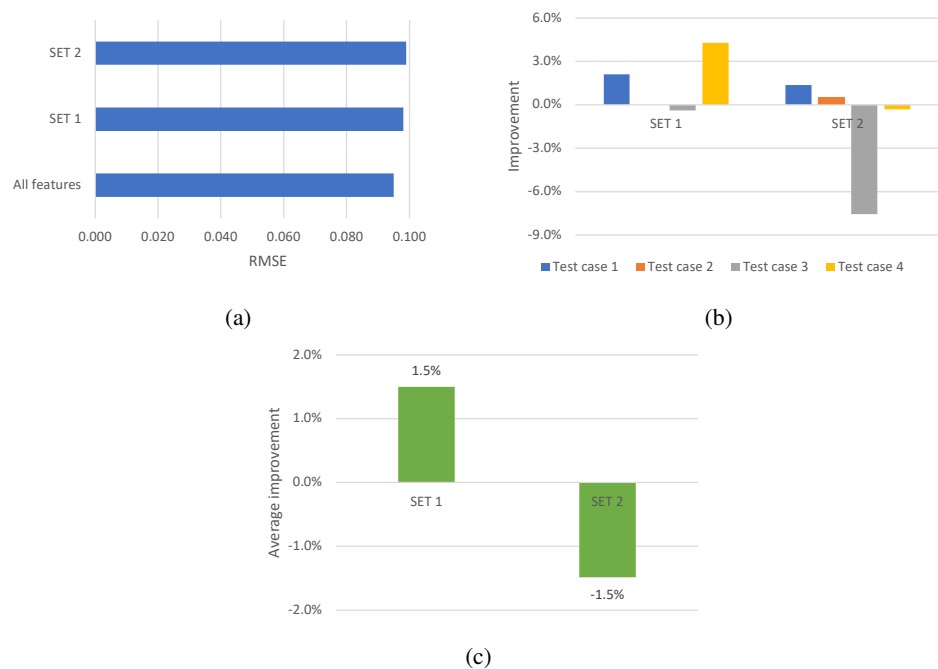

Figure 14: Comparison of different number of features. (a) the test root mean square error of the trained random forest models. (b) the improvement in the number on nonlinear iterations compared with the original random forest. (c) average improvement (reduction) in the number of nonlinear iterations compared with the original random forest.

iterations. The results show that the acceleration using the random forest with Set 1 performed better than the original random forest and the one with Set 2.

## F Machine learning model selection

The cumulative number of nonlinear iterations during the numerical simulation for all the machine learning models and the case with no relaxation can be seen in Figure 15.

## G Online training hyperparameters

For the xgboost with the boosting strategy and the random forest with the bagging strategy, we have tested different hyperparameters and the best results were achieved using the values in Tables 5 and 6, respectively.

Table 5: Hyperparameters used for the online training of the xgboost with the boosting strategy.

| PARAMETER | VALUE |
|---|---|
| NUMBER OF NEW BOOSTING TREES | 1 |
| LEARNING RATE | 0.01 |
| MAXIMUM DEPTH | 3 |
| SUBSAMPLE OF INSTANCES | 1.0 |
| SUBSAMPLE OF COLUMNS | 1.0 |
| BUFFER SIZE | 50 |

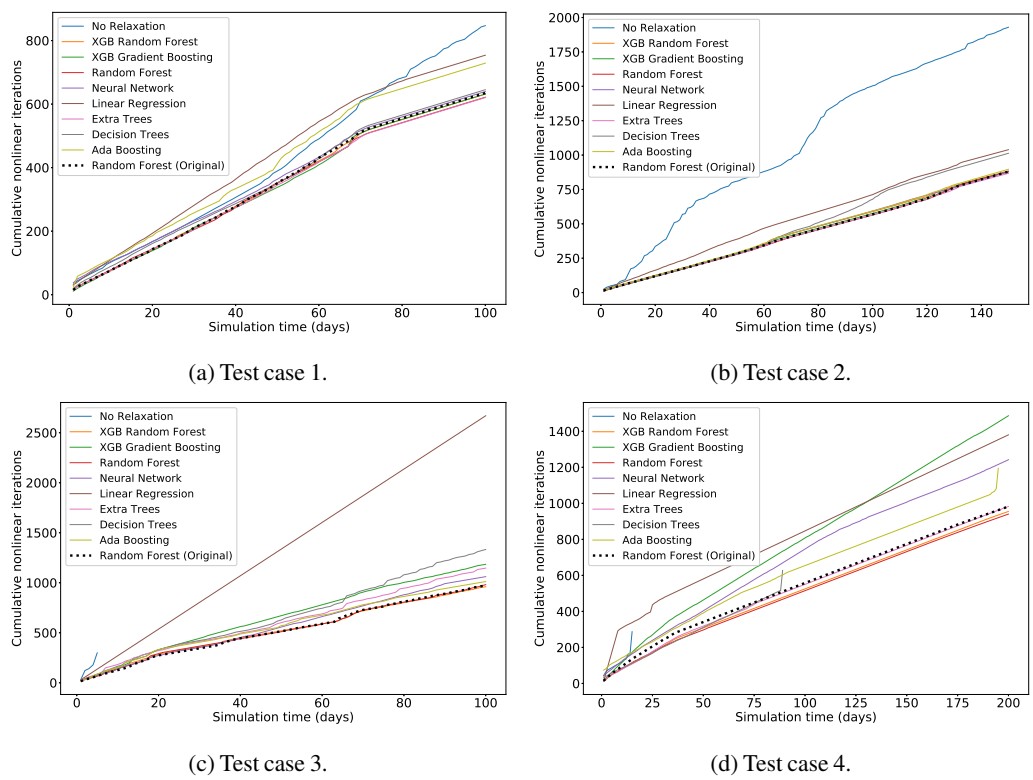

(a) Test case 1.        (b) Test case 2.

(c) Test case 3.        (d) Test case 4.

Figure 15: Cumulative number of nonlinear iterations for different machine learning models and for the case with no relaxation.

Table 6: Hyperparameters used for the online training of the random forest with the bagging strategy.

| PARAMETER | VALUE |
|---|---|
| NUMBER OF NEW TREES | 70 |
| MAXIMUM DEPTH | 30 |
| MAXIMUM NUMBER OF FEATURES | 0.2 |
| BUFFER SIZE | 50 |

# H   Hardware configuration

A Linux (Ubuntu 18.04.6 LTS) workstation was used to train the machine learning models and run all the numerical simulations. Table 7 shows the hardware configuration.

Table 7: Hardware configuration.

| Description | Quantity |
|---|---|
| 16GB DDR4 3200 MHz RAM Memory | 16 |
| Samsung 970 EVO PLUS 2TB SSD/Solid State Drive | 1 |
| Seagate IronWolf PRO 4TB SATA HDD/Hard Drive | 3 |
| Nvidia Quadro RTX 4000 Video Card | 1 |
| AMD 32 Core 2nd Gen EPYC 7452 CPU/Processor | 2 |
| AMD EPYC 7000 EATX Gigabit Server Motherboard | 1 |

