# OpenReview forum: "Adaptive learning acceleration for nonlinear PDE solvers"
_NeurIPS.cc/2023/Workshop/AI4Science — NeurIPS2023-AI4Science Poster_

### Official Review · Reviewer_3ivk · 2023-10-14
**A solid workshop paper.**

**Rating:** 7
**Confidence:** 4

**Review:**

This work uses online learning to accelerate the PDE solver and apply it to multiphase porous media problems. The paper is well-written and the results are encouraging. My main concern is whether the proposed approach is general enough to applied in other problems or designed specially for porous media. (It seems to me that the method is problem-specific.) Clarification on this issue in paper will be helpful.

---

### Official Review · Reviewer_kTRo · 2023-10-25
**Adaptive learning acceleration for nonlinear PDE solvers**

**Rating:** 6
**Confidence:** 3

**Review:**

**Summary:**
This paper proposes a nonlinear solver acceleration for nonlinear PDEs that is based on online learning, and it is applied on multiphase porous media flow.  This work identified the key dimensionless parameters required and showed the reduction in the number of nonlinear iterations obtained by using the proposed approach in complex realistic (three-dimensional) models. Additionally,  this work coupled a machine learning model into an open-source multiphase flow simulator achieving up to 85% reduction in computational time.

**Strengths and Weakness:**
- Overall this paper is well-written, and the presentation is clear and easy to follow.
- The approach is capable of reducing the number of nonlinear iterations by dynamically adjusting one single global parameter and by learning on-the-job the characteristics of each numerical model.
- The method is only evaluated on one system, I recommend using more PDEs to test the effectiveness of the proposed method.


**Limitations**:
The limitations of this paper is not well discussed.